# Prevalence of Intestinal Protozoa Among Patients Living with HIV in the Peruvian Amazon

**DOI:** 10.3390/tropicalmed10110324

**Published:** 2025-11-18

**Authors:** Silvia Otero-Rodriguez, Viviana Pinedo-Cancino, Martin Casapia-Morales, Victoria-Ysabel Villacorta-Pezo, Seyer Mego-Campos, Jorge Parráguez-de-la-Cruz, Esperanza Merino, Eva H. Clark, José-Manuel Ramos-Rincón

**Affiliations:** 1Infectious Diseases Unit, Doctor Balmis University General Hospital, 03010 Alicante, Spain; merino_luc@gva.es; 2Alicante Institute of Sanitary and Biomedical Research (ISABIAL), 03010 Alicante, Spain; jose.ramosr@umh.es; 3Faculty of Human Medicine, National University of the Peruvian Amazon, Iquitos 16007, Peru; viviana.pinedo@unapiquitos.edu.pe (V.P.-C.); mcasapia@acsaperu.org (M.C.-M.); 4Laboratory for Research on Natural Antiparasitic Products of the Amazon (LIPNAA-CIRNA), National University of the Peruvian Amazon, Iquitos 16007, Peru; megoseyer@gmail.com; 5Infectious Diseases and Tropical Medicine Service, Loreto Regional Hospital, Iquitos 16001, Peru; 6Medical Department, Asociación Civil Selva Amazónica, Iquitos 16001, Peru; 7Clinical Laboratory, National University of the Peruvian Amazon, Iquitos 16007, Peru; villacortapezovictoria@gmail.com; 8Clinical Laboratory, Asociación Civil Selva Amazónica, Iquitos 16001, Peru; jparraguez@acsaperu.org; 9Clinical Medicine Department, Miguel Hernández University of Elche, 03202 Elche, Spain; 10Department of Medicine (Infectious Diseases) and Department of Pediatrics (Tropical Medicine), Baylor College of Medicine, Houston, TX 77030, USA; 11Internal Medicine Department, Doctor Balmis University General Hospital, 03010 Alicante, Spain

**Keywords:** intestinal protozoa, parasite, *Cryptosporidium*, *Giardia*, *Entamoeba*, HIV, Peru

## Abstract

Intestinal protozoa are a common cause of morbidity in people living with HIV (PWH), particularly in tropical regions with poor sanitation. We conducted a cross-sectional study in 315 PWH from Iquitos, Peru, between October 2023 and May 2024, to assess their prevalence and risk factors. Stool samples were examined using Lugol’s iodine, modified Ziehl–Neelsen (MZN) staining, and immunochromatography (ICT). The mean age was 41 years, with a median CD4+ count of 431 cells/µL; 12.4% were in the AIDS stage, and 21.5% had a detectable viral load. 51.4% of the participants tested positive for any intestinal protozoa. The overall *Cryptosporidium* spp. prevalence (by combining MZN and ICT results) was 25.7%. The overall *Giardia* spp. and *Entamoeba* spp. prevalences (by combining Lugol’s iodine and ICT results) were 2.9% and 1.9%, respectively. *Blastocystis* spp. was frequently isolated, though its pathogenicity remains uncertain. Diagnostic agreement was almost perfect between Lugol and ICT for *Giardia* and *Entamoeba* (κ = 0.87; *p* < 0.001 and κ = 0.91; *p* < 0.001, respectively), but only slight between MZN and ICT. Homosexual practices were identified as a significant risk factor for pathogenic protozoa infection (AOR 2.52; 95% CI: 1.04–6.12). In conclusion, the high prevalence of protozoa infection reflects ongoing fecal–oral exposure, underscoring the need for public health education, routine diagnosis, and treatment in similar settings.

## 1. Introduction

Intestinal protozoa are common in immunocompromised patients and an important cause of diarrhea in patients living with HIV (PWH) worldwide [1], particularly in tropical low-resource countries that are challenged by scarce access to potable water and/or robust sanitation infrastructure [2,3].

*Cryptosporidium* spp. is a well-recognized opportunistic intestinal protozoa that cause acute or chronic enterocolitis, with severity varying according to the host’s immune status [4]. In fact, it is classified as an acquired immunodeficiency syndrome (AIDS)-defining illness, as it predominantly affects PWH with profound immunosuppression, in whom it can lead to severe, life-threatening diarrheal disease [4,5,6,7]. Additionally, it can cause epidemic outbreaks [8]. *Giardia duodenalis* can cause similar gastrointestinal manifestations, particularly diarrhea and malabsorption, which can be more persistent and difficult to treat in immunocompromised hosts, although it is less frequently associated with life-threatening disease [9]. *Entamoeba histolytica* is a well-recognized pathogenic intestinal protozoan capable of causing amebic colitis, with disease severity largely determined by the host’s immune status. In immunocompromised patients, such as those with advanced HIV infection, *E. histolytica* infection may progress to more severe manifestations, including amebic liver abscesses and fulminant amebic dysentery [10].

*Blastocystis* spp., traditionally known as a commensal parasite, is a parasite frequently found in human fecal samples [11,12]. Its pathogenicity remains uncertain; however, certain subtypes (such as ST1 and ST7) have been associated with gastrointestinal symptoms—including diarrhea, bloating, or abdominal pain—and, in some reports, with irritable bowel syndrome and pruritus, possibly mediated through alterations of the intestinal microbiome [13]. This suggests that the clinical relevance of *Blastocystis* may be limited to specific populations and clinical contexts. *Entamoeba coli*, *Endolimax nana*, and *Iodamoeba bütschlii* are generally considered non-pathogenic in humans, but their presence in stool may serve as a sentinel for infection with pathogenic organisms [14].

The prevalence of intestinal protozoa is likely underestimated given the high proportion of asymptomatic infections, low sensitivity of standard microscopy, and scarcity of personnel who are competent in parasitology diagnosis [6]. Despite this, it is consistently reported to be high among PWH in Peru [15], sometimes reaching levels twice as high as in healthy individuals [16]. Some reports describe intestinal protozoa in PWH in Lima, with an overall prevalence of 47.5%. *Cryptosporidium* spp., *Blastocystis* spp., and *Giardia duodenalis* were the most common, with prevalences of approximately 20%, 11%, and 8%, respectively [17,18]. However, no studies have assessed the HIV population in the Peruvian Amazon basin.

Iquitos, Peru, the largest city in the Peruvian Amazon (Loreto Department), presents a unique context for studying the prevalence of intestinal protozoa in PWH. It has the second-highest prevalence of PWH in Peru, after Lima; more than 1000 patients presented to Iquitos’ public hospitals, and 272 new HIV cases were diagnosed in the latter part of 2023 [19]. The city has distinctive environmental and socio-economic factors (hot and humid conditions, limited access to healthcare and potable water, high levels of poverty), which could contribute to a high burden of intestinal parasitic infections. Despite the favorable environment for intestinal protozoa infections, their epidemiology in PWH is understudied. Addressing this knowledge gap is crucial for developing targeted clinical and public health interventions to improve the care of PWH.

The objective of this study was to assess the parasitological prevalence and risk factors for intestinal protozoa infection in established outpatients attending HIV-dedicated clinics in Iquitos, Peru.

## 2. Materials and Methods

We conducted a cross-sectional study of PWH receiving care at one of two hospitals in Iquitos, Loreto Department, Peru: (1) the Regional Hospital of Loreto “Felipe Santiago Arriola Iglesias”, a referral center for patients from the northern part of the city or surrounding rural communities), and (2) the Hospital of Iquitos “César Garayar García”, a referral center for patients from the southern part of the city, from 20 October 2023 to 20 May 2024.

### 2.1. Study Population and Inclusion/Exclusion Criteria

We included adult outpatients (≥18 years) with confirmed HIV infection who were attending HIV-dedicated clinics for routine follow-up at either the Regional Hospital of Loreto or the Hospital of Iquitos and were able to provide stool specimens. Previously enrolled patients were excluded from re-entry during the study period.

### 2.2. Enrollment Procedures

Patients were enrolled consecutively after being informed about the study when they attended either hospital for their routine follow-up visits, which are typically scheduled every six months. After informed consent, the participant’s socio-epidemiological, clinical, and HIV-related data were collected through a structured interview and recorded in an electronic Excel spreadsheet due to the lack of a stable internet connection. The presence of diarrhea was classified according to the patient’s self-report, while frequency was categorized into three categories (at least once a day, at least once a week, at least once a month). Next, study participants submitted a fresh stool specimen, which was preserved under cold-chain conditions and processed for copro-parasitological examination on the same day. If participants could not submit the stool sample, a driver collected it from their home within the following week, provided they consented to a home visit. All participants agreed to home collection. The driver was available by call from 9:00 a.m. to 7:00 p.m., Monday through Friday, with patients instructed to keep samples refrigerated as much as possible; transport to the clinic usually took 20–30 min, during which samples were maintained in a cooler to preserve the cold chain.

### 2.3. Stool Specimen Processing, Staining, and Microscopy

Stool samples were examined using Lugol’s iodine solution for the identification of intestinal protozoa; modified Ziehl–Neelsen (MZN) stain for *Cryptosporidium* spp. oocysts (a technique not routinely performed in Iquitos); and an immunochromatographic test (ICT) for *Cryptosporidium* spp., *Giardia duodenalis*, and *Entamoeba histolytica*/*dispar*. All the positive samples were assessed by two technicians, together with the 20% negative stool (control quality) and the MZN stains. Discordant results were re-evaluated until consensus was reached. In addition, standard quality control procedures were followed throughout the staining and reading processes.

-Lugol’s iodine solution: Each fecal specimen was analyzed using Lugol’s iodine solution to enhance the diagnostic accuracy of direct microscopy of wet mounts, looking for *Giardia* spp., *Entamoeba* spp., *Blastocystis* spp., *Entamoeba coli*, *Endolimax nana*, and *Iodamoeba bütschlii.* Due to the expected high volume of samples and the labor-intensive nature of some techniques, a concentration method was not performed. Moreover, given the anticipated high prevalence of intestinal protozoa in this population and the use of additional diagnostic methods (ICT and MZN), the benefit of performing a concentration technique was considered limited [20]. Lugol’s iodine stains glycogen and other cytoplasmic structures, enhancing the visualization of protozoan cysts and trophozoites. *Giardia* cysts typically appear oval with internal nuclei and axonemes [21], while *Entamoeba* cysts show characteristic nuclear structures and chromatoid bodies, and the trophozoite could appear with red blood cells in the cytoplasm, which allows it to be distinguished from the commensal *E. dispar*, even if it is not a pathognomonic sign [22]. Commensal protozoa, including *Blastocystis*, display variable shapes and internal granularity, whereas *E. coli*, *E. nana*, and *I. bütschlii* cysts can be distinguished by their size, number of nuclei, and cytoplasmic inclusions [23]. This technique is simple, fast, and useful, and provides a cost-effective approach for preliminary identification of intestinal protozoa in laboratory settings.-Modified Ziehl–Neelsen stain (MZN): Briefly, each stool sample was homogenized, and a thin smear was prepared on a slide. After heat-fixing, slides were stained with phenolated fuchsin, decolorized with 3% acid alcohol, counterstained with methylene blue, air-dried, and observed under immersion oil at 100× magnification [24]. To assure high-quality microscopy results, the two study staff microscopists were trained by the Cayetano Heredia University’s Microbiology Service and Selva Amazonica Civil Association before study initiation.-*Crypto + Giardia + Entamoeba* ICT (CerTest^®®^, Certest Biotec, Zaragoza, Spain) [25]: This one-step combo card test is a colored chromatographic immunoassay for the simultaneous qualitative detection of *Cryptosporidium* spp. (via Anti-Crypto MAb (clone CR23) and inactivated *Cryptosporidium parvum* antigen (native extract)), *Giardia duodenalis* (via α1-giardin protein and/or the cyst wall protein CWP1, detecting both trophozoites and cysts) and *Entamoeba histolytica*/*dispar* (via antigens from both species) in stool samples [26]. It is used by mixing a small amount of stool sample with the provided buffer, applying the mixture to the test cassette, and waiting the specified time (usually 10–15 min). The appearance of lines in the result window indicates the presence of antigens from *Cryptosporidium* spp. and/or *Giardia duodenalis* and/or *Entamoeba histolytica*/*dispar*.

### 2.4. Data Analysis

Statistical analyses were performed via SPSS Statistics version 22.0 (IBM, Armonk, NY, USA). For descriptive statistics, categorical variables are expressed as frequencies and percentages, while continuous variables are presented as medians with interquartile range (IQRs). The 95% confidence intervals (CIs) were calculated using the Newcombe method [27]. Categorical variables were compared using Chi-square tests, while continuous variables were analyzed using Student’s *t*-tests (for variables with a normal distribution, like age) and Mann–Whitney U tests (for other quantitative variables without a normal distribution). The prevalence of each parasite was reported separately for each diagnostic method (Lugol’s microscopy, ICT, and MZN staining), as well as an overall prevalence for *Giardia* spp., *Entamoeba histolytica*/*dispar*, and *Cryptosporidium* spp., defined as follows:Overall *Giardia* spp. prevalence: combined positivity in Lugol’s microscopy and ICT;Overall *Entamoeba histolytica*/*dispar* prevalence: combined positivity in Lugol’s microscopy and ICT;Overall *Cryptosporidium* spp. prevalence: combined positivity in MZN staining and ICT.

Finally, we also present the prevalence in the subgroup of patients reporting diarrhea.

To compare the agreement between two diagnostic tests (MZN versus ICT for *Cryptosporidium* spp. and Lugol’s iodine solution versus ICT for the other two protozoa), we calculated Cohen’s kappa coefficient, which measures concordance beyond chance. Kappa values were interpreted as slight (0–0.20), fair (0.21–0.40), moderate (0.41–0.60), substantial (0.61–0.80), or almost perfect (0.81–1.00) agreement. McNemar’s test for paired proportions was performed to evaluate whether there were significant differences in the number of positive results detected by each method.

Risk factors were assessed for the overall prevalence of the three main pathogenic intestinal protozoa (*Cryptosporidium*, *Giardia*, and *Entamoeba*). Initially, we evaluated them through bivariate analysis, with associations quantified using odds ratios (ORs). Subsequently, multivariable logistic regression models were constructed to identify independent risk factors for protozoan infection. This model included variables that showed statistical significance (*p* < 0.05) in the univariate analyses, adjusted for gender and age. The models’ goodness of fit was assessed using Cox–Snell R^2^ and Nagelkerke R^2^ statistics to evaluate the association between the dependent variable (protozoa infection) and independent variables (socio-epidemiological, clinical, and HIV-related factors).

### 2.5. Ethical Considerations

The Ethics Committee of Loreto Regional Hospital in Iquitos (Peru) (EXP: ID-018-CIEI-2013) and the Responsible Research Office of the Miguel Hernández University of Elche approved the study (DMC.JMRR.230908). All participants provided written informed consent. All study results were kept strictly confidential and released only to the participants’ HIV healthcare providers, who offered treatment and follow-up to those who tested positive for intestinal parasites (protozoa or helminths) at no cost.

## 3. Results

### 3.1. Description of the Cohort

315 patients were enrolled (Figure 1). The mean age of the PWH cohort was 41 (+/−11 years), with a median of CD4+ count of 431 cells/µL (IQR 288, 584). 39 patients (12.4%) had CD4+ count < 200 cells/µL (AIDS stage), while 68 (21.5%) had a detectable viral load.

Most participants (267/315, 85%) were enrolled from Loreto Regional Hospital, which has the largest PWH cohort in Iquitos. Most participants were heterosexual men (227/315, 76.7%) and had few comorbidities. 103/315 (32.7%) resided on an unpaved road, 150/315 (47.6%) lived in a wood-made house, and 220/315 (70.0%) reported frequent contact with domestic animals. Of those reporting diarrhea (68/315, 21.6%), 76.5% reported symptoms once a month, 10.3% once a week, and 13.2% once a day. Baseline characteristics remained similar after excluding the 41 patients with incomplete data (n = 356 → 315).

### 3.2. Stool Diagnosis

We received stool samples from 315 PWH. All samples underwent direct examination using Lugol’s iodine staining, MZN, and ICT. Among the 315 samples, 162 (51.4%; 95% CI 45.9–57.0%) were positive for any pathogenic or commensal protozoa.

#### 3.2.1. Prevalence of *Giardia* spp., *Entamoeba* spp., *Blastocystis* spp., and Commensal Pathogens

By Lugol’s iodine staining, 35/315 (prevalence 11.1%; 95% CI 8.1–15.1%) were positive for *Blastocystis* spp., 8/315 (prevalence 2.5%; 95% CI 1.2–4.9%) for *Giardia* spp., and 5/315 (prevalence 1.6%; 95% CI 0.7–3.7%) for *Entamoeba* spp. For commensal pathogens, 64/315 (prevalence 20.3%; 95% CI 16.3–25.1%) were positive for *Entamoeba coli*, 24/315 (prevalence 7.6%; 95% CI 5.2–11.1%) for *Endolimax nana*, and 6/315 (prevalence 1.9%; 95% CI 0.8–4.1%) for *Iodamoeba buetschlii*.

By ICT, 8/315 (prevalence 2.5%; 95% CI 1.2–4.9%) were positive for *Giardia duodenalis*, and 6/315 (prevalence 1.9%; 95% CI 0.8–4.1%) for *Entamoeba histolytica*/*dispar.*

By combined methods (overall prevalence): 9/315 (prevalence 2.9%; 95% CI 1.5–5.3%) were positive for *Giardia* spp., and 6/315 (prevalence 1.9%; 95% CI 0.8–4.1%) for *Entamoeba* spp.

#### 3.2.2. Prevalence of *Cryptosporidium* spp.

By MZN, 73/315 (prevalence 23.2%; 95% CI 18.9–28.1%) were positive for *Cryptosporidium* spp.

By ICT, 11/315 (prevalence 3.5%; 95% CI 2.0–6.1%) were positive for *Cryptosporidium* spp.

By combined methods (overall prevalence), 81/315 (prevalence 25.7%; 95% CI 21.2–30.8%) were positive for *Cryptosporidium* spp.

#### 3.2.3. Evaluation of Diagnostic Test Agreement

Agreement between Lugol’s iodine microscopy and ICT for the detection of *Giardia* spp. and *Entamoeba* spp. was almost perfect (κ = 0.87; 95% CI 0.70–1.00; *p* < 0.001 and κ = 0.91; 95% CI 0.73–1.00; *p* < 0.001, respectively), while McNemar’s test did not indicate a significant difference between discordant pairs (*p* = 1.000) in both cases. In contrast, agreement between MZN and ICT for the detection of *Cryptosporidium* spp. was slight (κ = 0.11; 95% CI 0.04–0.18; *p* = 0.74), while McNemar’s test revealed a significant difference in discordant pairs (*p* < 0.001) (Figure 2).

#### 3.2.4. Prevalence of Co-Infection with *Giardia* spp., *Entamoeba* spp., *Cryptosporidium* spp., and *Blastocystis* spp.

8/315 patients (2.5%; 95% CI 1.3–4.9) were co-infected with *Cryptosporidium* spp. and *Blastocystis* spp., 3/315 (1.0%; 95% CI 0.3–2.9) with *Cryptosporidium* spp. and *Giardia* spp., and 2/315 (0.6%; 95% CI 0.2–2.3) with *Giardia* spp. and *Blastocystis* spp.

#### 3.2.5. Epidemiological Risk Factors Associated with Pathogenic Intestinal Protozoa Positivity

We compared demographic and epidemiological characteristics, comorbidities, and HIV infection features by pathogenic protozoa (overall *Cryptosporidium* spp., *Giardia* spp., and *Entamoeba* spp. positivity) (Table 1).

After adjusting for sex, age, and variables with *p*-values < 0.10 in the bivariate analysis (hospital, current CD4+ count < 200 cells/μL, viral load uncontrolled >20 copies/mL, daily diarrhea, homosexual practices, prior intestinal parasite infection, gonococcal infection and toxoplasmosis), homosexual practices (compared with heterosexual, bisexual, or transgender practices) were significantly associated with a higher risk of pathogenic protozoa positivity (*p*: 0.045; adjusted OR: 2.52; 95% CI: 1.04–6.12). As including CD4+ count in the multivariable model reduced the analytic sample to 199 participants (63.2% of the total), due to missing data for this variable, an additional multivariable analysis was performed excluding CD4+, which increased the sample size to 285 participants (90.5% of the cohort). The second model yielded consistent results, showing that homosexual risk practices were also associated with a higher prevalence of intestinal parasitosis (*p*: 0.028; adjusted OR: 2.32; 95% CI: 1.10–4.91), together with a history of previous gonococcal infection (*p*: 0.022; adjusted OR: 2.77; 95% CI: 1.16–6.59).

#### 3.2.6. Prevalence Pattern of Protozoa in People Referring Diarrhea

Among patients with diarrhea, the prevalence pattern of pathogenic agents was higher than in asymptomatic individuals. *Cryptosporidium* spp. was the most frequently isolated parasite (19/68, 27.9%; 95% CI 18.7–40.0), followed by *Blastocystis* spp. (8/68, 11.8%; 95% CI 6.1–21.5), *Giardia* spp. (4/68, 5.9%; 95% CI 2.3–14.2), and *Entamoeba* spp./3/68, 4.4%; 95% CI 1.1–13.1).

The presence of overall *Giardia* spp. was marginally associated with diarrhea in the previous month (*p* = 0.091; OR 3.03; 95% CI 0.79–11), which was similar for *Entamoeba* spp. (*p* = 0.088; OR 3.754; IC 0.74–19.04)), without significant differences in the sub-analysis by Lugol’s iodine or ICT techniques. A prior diagnosis of gonorrhea was significantly associated with diarrhea (*p* = 0.012; OR 2.41; 95% CI 1.20–4.86), as was a history of digestive disease (*p* < 0.001; OR 4.09; 95% CI 1.63–10.28).

## 4. Discussion

This study provides an epidemiologic evaluation of intestinal protozoa infections among a vulnerable population—people with HIV (PWH)—in Iquitos, Peru, representing one of the few studies conducted in a tropical rainforest environment, where ecological conditions for the transmission and acquisition of intestinal parasites are particularly favorable. The global prevalence of intestinal parasitosis in patients with HIV/AIDS and diarrhea has been reported at approximately 50%, with various studies identifying *Cryptosporidium* spp. as the most common protozoan pathogen, followed by *Giardia* spp. [16,17,28], which is consistent with our results.

### 4.1. Cryptosporidium spp. Prevalence in Stool

Our study demonstrated a notable prevalence of *Cryptosporidium* spp. in both the general cohort (25.7%) and people with diarrhea (27.9%), relatively high compared to reports from similar populations in Peru, where prevalence usually ranges between 10 and 20% depending on diagnostic methods and setting [16,17,18,29], reaching 25% among PWH with poor hygienic habits [30]. We found only one study of *Cryptosporidium* spp. in Amazonian populations without HIV [31], in children living in the Colombian Amazon, which identified a 2% prevalence, suggesting that our results are among the first to describe *Cryptosporidium* infections in PWH in the Peruvian Amazon.

The substantial discrepancy between MZN (23.2%) and ICT (3.5%) results, including 70 MZN-positive/ICT-negative samples, represents a key finding and a central point of discussion in this study. Our results can be attributed to two main factors: First, MZN is a specific tool for detecting *Cryptosporidium* spp., but its sensitivity can vary (60–100%) depending on the staining process, infection stage, and the laboratorian’s expertise [8,32]. The presence of other acid-resistant or refractory materials in fecal samples (e.g., *Cystoisospora* or *Cyclospora cayetanensis* oocysts, yeast cells, pollen, or fungal spores) may cause what appears to be a Cryptosporidium oocyst to actually be debris or a different microorganism, resulting in false-positive results [33,34]. A minor potential for misidentification may persist even when samples are evaluated independently by two expert microscopists, as was the case in our study. Second, while MZN detects oocysts from any *Cryptosporidium* species, the manufacturer’s instructions for the CerTest “Crypto + Giardia + Entamoeba” targets have ambiguities in the specification for the *Cryptosporidium parvum*/*Cryptosporidium* hominis module in their rapid antigen tests. The combined “Crypto + Giardia + Entamoeba” test [25] lists *Cryptosporidium* generically, whereas the individual CerTest Crypto Card [35] explicitly targets *C. parvum.* When reviewing the list of raw materials, we found that the test includes a pan-specific monoclonal antibody (clone CR23) as well as an inactivated *C. parvum* antigen [26]. As a result, although the test is marketed for *Cryptosporidium* spp., its validation appears focused on *C. parvum*, leaving uncertainty regarding the detection of *C. hominis* and raising the potential for underdiagnosis in regions where *C. hominis* predominates. We only found an external validation of the individual CerTest Crypto Card in Sub-Saharan Africa, with a good specificity (92.5%) but low sensitivity (49.6%) and positive predictive value (61.3%) when compared to the composite reference standard of qPCR and RFLP-PCR for the detection of *Cryptosporidium* species [36]. While *Cryptosporidium parvum* is generally considered the most relevant species in low-resource and rural settings due to its zoonotic transmission—accounting for up to 70% of infections in some HIV-positive cohorts in Asia [37]—studies among HIV-infected patients in Lima reported that *C. parvum* represented only 11.3% of identified genotypes, whereas *C. hominis* (67.5%) and *C. meleagridis* (12.6%) were predominant [38]. Although species distribution data from Iquitos are lacking, the low ICT positivity may indicate a predominance of non-*parvum* species. Further validation studies are needed to accurately evaluate the ICT performance across diverse epidemiological settings.

With the available data, the high prevalence of *Cryptosporidium* spp. detected by the modified Ziehl–Neelsen (MZN) technique may represent the true burden of infection in this setting. Nevertheless, as MZN has lower specificity than molecular assays, the possibility of overdiagnosis cannot be entirely ruled out without PCR confirmation. Future studies incorporating molecular diagnostics are warranted to validate these findings.

### 4.2. Giardia and Entamoeba spp. Prevalence in Stool

Prevalence of *Giardia* spp. and *Entamoeba* spp. was relatively low, but agreement between Lugol’s iodine microscopy and ICT for the detection of *Giardia duodenalis* and *Entamoeba histolytica*/*dispar* was almost perfect, underscoring the validity of the diagnostic approach [20]. The prevalence of *Giardia* spp. observed in this study is notably lower than global estimates. A meta-analysis in HIV/AIDS patients reported a pooled prevalence of giardiasis of 5% [39]. Studies in Peru have reported considerable variability depending on population, area, and associated symptomatology, with prevalence rates as high as 20% in Trujillo (northern Peru) [40] and 4–15% in HIV in Lima [18,41]. In the Peruvian Amazon, studies on intestinal protozoa are limited. One study found high prevalences at both the Military and Regional Hospitals of Loreto in Iquitos, though higher in the former (15% vs. 4.8%) [42]. Additional reports from rural areas such as Yurimaguas [43] and native communities in the upper Marañón (Amazonas) [44] have documented similarly elevated prevalences, ranging from 17% to 21%. One possible explanation is that the presence of other opportunistic microorganisms, such as *Cryptosporidium*, could compete with and displace *Giardia*, especially in individuals with more advanced immunosuppression [45]. Furthermore, *Cryptosporidium* oocysts are more resistant to conventional cleaning methods than *Giardia* oocysts, and they can be more easily acquired in recreational settings such as swimming pools and rivers, which are very common in Iquitos [46]. Finally, children, who were not included in our study, appear more susceptible to *Giardia* infection [47].

*Entamoeba histolytica* is usually less frequent than *Giardia duodenalis*, but its prevalence may be overestimated because it is difficult to distinguish pathogenic *E. histolytica* from the non-pathogenic *E. dispar*. Few studies in Peru have assessed its prevalence. One study in Lima reported a prevalence of 1.9% among people living with HIV [41], consistent with our results. In two studies in Iquitos in non-HIV people, prevalence was 4.8% in Loreto Regional Hospital, 10% in the Militar Hospital [42], and 13% in a rural location in the Marañón River [44].

Although the associations between the presence of *Giardia* or *Entamoeba* and diarrhea were not statistically significant, our results suggest a trend toward an increased risk. However, the occurrence of diarrhea in this population is likely multifactorial. A prior diagnosis of gonorrhea and a history of gastrointestinal disease—both significantly associated with diarrhea—support the influence of other contributing factors. Beyond enteric protozoa, episodes of diarrhea among PWH in Iquitos may also be driven by other coinfections (*Campylobacter* spp., *Escherichia coli*…), anti-retroviral-related gastrointestinal effects, or changes in gut microbiota related to HIV infection [16,48]. In addition, the local diet, typically fried and rich in fats and calories, may predispose individuals to postprandial gastrointestinal discomfort or diarrhea, potentially affecting the specificity of symptom-based associations [49].

### 4.3. Prevalence of Blastocystis and Commensal Pathogens in Stool

*Blastocystis* spp. prevalence was high among PWH in Iquitos (11.5%). Globally, estimates vary from 10 to 50% depending on the geographical area [50]. In Peru, studies in PWH are mostly restricted to Lima, reporting prevalences between 11% [18] and 24.6% [41], consistent with our findings. In our study, *Blastocystis* spp. was not associated with diarrhea but was identified as a risk factor for co-infection with commensal protozoa, similar to previous reports suggesting that *Blastocystis* may serve as a sentinel of fecal contamination [14] due to its association with limited access to potable water and animal contact [50,51,52].

### 4.4. Risk Factors for Pathogenic Intestinal Protozoa Acquisition

PWH are well-known to be a vulnerable population for intestinal parasitosis, particularly opportunistic protozoa such as *Cryptosporidium* spp. in patients with low CD4+ counts [53]. Additional commonly reported risk factors for intestinal protozoan infections include young age, male sex, low educational level, lack of sanitary facilities, previous infection with other protozoa, living in suburban areas, and uncontrolled HIV viral load (>1000 copies/mL) [53,54].

In our study, homosexual practices were associated with a higher prevalence of pathogenic protozoa. This finding aligns with previous research describing the role of specific sexual behaviors (particularly oral–anal contact (“rimming”)), digital–anal contact, use of sex toys contaminated with fecal material or multiple sexual practices without intermediate hygiene) as potential routes for fecal–oral transmission of enteric parasites [55,56]. These include *Giardia* spp., *Entamoeba* spp., *Cryptosporidium* spp., *Blastocystis* spp., and some helminths such as *Strongyloides stercoralis* [57,58,59,60,61]. Furthermore, HIV infection may amplify the risk of enteric pathogen acquisition during sexual activity by compromising mucosal immunity and altering gut integrity [56]. In the multivariable model excluding CD4+ count, this association was confirmed, and a previous gonococcal infection also remained independently associated with intestinal protozoa infection, reinforcing the link between sexual risk behaviors and intestinal parasitic infections.

Our findings suggest that intestinal protozoa may share transmission routes with HIV, emphasizing the importance of prevention strategies that integrate education on sexual health, hygiene, and awareness of enteric pathogens in populations engaging in high-risk sexual behaviors.

Furthermore, a poor immunovirological control of HIV and a previous diagnosis of toxoplasmosis—which also reflects impaired immunity—were identified as risk factors for protozoan acquisition in our bivariate analysis [17,62,63], even if they did not emerge as risk factors in the multivariable analysis. This was likely due to the high proportion of missing HIV viral load data, which substantially reduced the number of participants included. Other important associations, such as a prior diagnosis of intestinal parasites or experiencing daily diarrhea, also reached significance in the bivariate analysis but not in the multivariate analysis, suggesting trends that could be confirmed with larger cohorts.

### 4.5. Strengths and Limitations

This study is the first epidemiological evaluation of intestinal protozoa—including *Cryptosporidium* spp., *Giardia* spp., and *Entamoeba* spp. —in PWH in the Peruvian Amazon. It identifies risk factors for protozoan acquisition in this vulnerable population and may inform clinical and public health interventions. Strengths include rigorous staff training in stool processing and microscopy, use of gold-standard techniques for *Cryptosporidium* detection, and inclusion of participants from both major hospitals in Iquitos, representing about one-third of the city’s PWH.

Nevertheless, this study is subject to several limitations. First, a stool concentration technique, which may have improved diagnostic sensitivity, was not performed. However, their practical advantage may be reduced in contexts where parasite loads are high, samples are fresh, microscopy is performed by trained personnel, or complementary methods—such as ICT and MZN—are used, as in this study [20]. Second, resource limitations precluded the combination of traditional staining with newer molecular techniques (PCR), which would have improved the specificity and sensitivity of our results [34]. In this regard, it was not possible to distinguish *E. histolytica* from *E. dispar,* nor to differentiate the genotype of *Cryptosporidium* spp. A third limitation related to diagnostics is the potential for misclassification bias resulting from the discrepancy between MZN and ICT results. Future studies should incorporate PCR and genotyping approaches to accurately identify protozoan species and genotypes, which would allow more precise prevalence estimates and a better understanding of epidemiological patterns. Fourth, we faced a substantial amount of missing data for CD4 counts and HIV viral load, which limited the scope of our bivariate analyses and reduced the statistical power of the multivariable analysis. However, similar findings were obtained when this variable was excluded from the multivariable model, suggesting that the main associations observed are robust despite the reduced dataset. Finally, the cross-sectional design, which precludes causal inference, the exclusion of out-of-care PWH—which may have led to an underestimation of intestinal protozoa prevalence in the broader population—and potential recall or reporting bias for diarrhea and exposure history are additional limitations of our study. Our findings may not be generalizable to regions with different epidemiological profiles.

## 5. Conclusions

Our study demonstrates a higher-than-expected prevalence of *Cryptosporidium* spp. infection among PWH in Iquitos, affecting nearly one in four participants, whereas *Giardia* spp. and *Entamoeba* spp. were less common. Additionally, individuals reporting homosexual practices had an increased risk of acquiring pathogenic protozoa. These findings underscore the importance of implementing affordable laboratory techniques, such as Lugol’s iodine staining and MZN, to enable accurate screening for intestinal protozoal infections, particularly in patients with poor immunovirological control of HIV.

## Figures and Tables

**Figure 1 tropicalmed-10-00324-f001:**
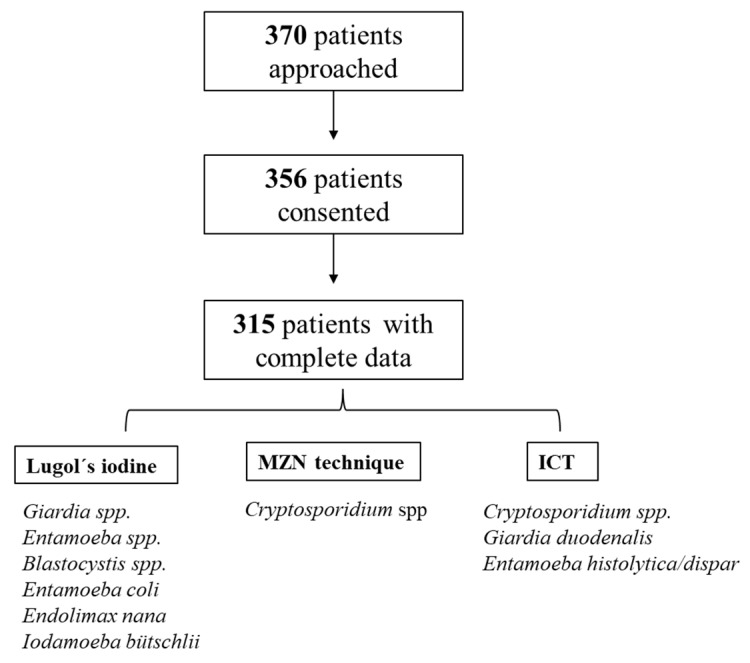
Flowchart illustrating sample availability for the study.

**Figure 2 tropicalmed-10-00324-f002:**
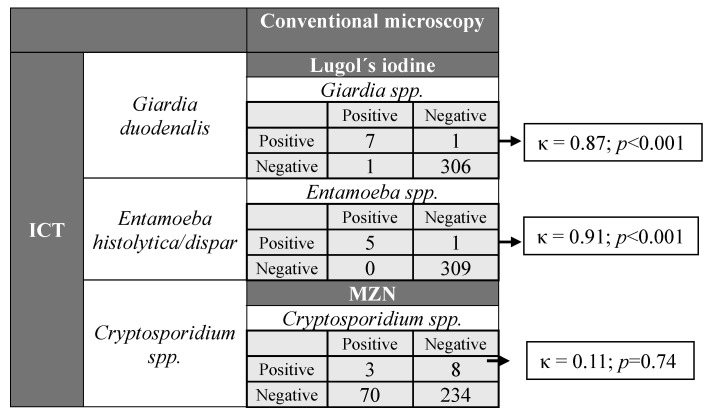
Cross-tabulations and Cohen’s Kappa (κ) for ICT versus conventional microscopy for *Giardia* spp., *Entamoeba* spp., and *Cryptosporidium* spp. Kappa values were interpreted as follows: slight (0–0.20), fair (0.21–0.40), moderate (0.41–0.60), substantial (0.61–0.80), and almost perfect (0.81–1.00) agreement.

**Table 1 tropicalmed-10-00324-t001:** Bivariate analysis of epidemiological characteristics of study participants (N = 315) by pathogenic protozoa positivity (overall *Cryptosporidium* spp., *Giardia* spp., and *Entamoeba* spp. positivity).

**Variables**	**Pathogenic** **Protozoa** **Positive (N = 92)**	**Pathogenic** **Protozoa** **Negative (N = 223)**	**OR**	** *p* **
**Male**, % (***n/N***)	60.9 (56/92)	64.6 (144/223)	0.85	0.54
**Age, mean ± SD, years**	42 ± 12	41 ± 12	1.26	0.39
**Hospital attended**, % (***n/N***)				
** Hospital of Iquitos**	**26.1 (24/92)**	**10.8 (24/223)**	**2.93**	**<0.001**
Regional Hospital of Loreto	73.9 (68/92)	89.2 (199/223)		
**Residence**, % (***n/N***)				
Iquitos district	33.7 (31/92)	34.5 (77/223)	N/A	0.58
Punchana district	29.3 (27/92)	25.1 (56/223)		
Belen district	15.2 (14/92)	15.7 (35/223)		
San Juan district	16.3 (15/92)	22.0 (49/223)		
Outside of Iquitos	5.4 (5/92)	2.7 (6/223)		
**Occupation**, % (***n/N***)				
** Unemployed or student (yes)**	38.1 (35/92)	46.2 (103/223)	N/A	0.19
Cattle, agriculture or construction (yes)	16.3 (15/92)	15.2 (34/223)		
Craft work (yes)	4.3 (4/92)	4.5 (10/223)		
Intellectual work ^a^ (yes)	15.2 (14/92)	6.7 (15/223)		
Self-employment (yes)	26.1 (24/92)	27.4 (61/223)		
**Education**, % (***n/N***)				
None (yes)	3.3 (3/92)	2.7 (6/223)	N/A	0.31
Attended primary school (yes)	22.2 (20/92)	14.3 (32/223)		
Attended secondary school (yes)	48.9 (45/92)	59.2 (132/223)		
Attended university (yes)	26.1 (24/92)	23.8 (53/223)		
**Epidemiological risk factors, % (*n/N*)**				
Lives with dogs/cats/farm animals (yes)	69.6 (64/92)	70.0 (156/223)	0.98	0.94
Walks barefoot (yes)	33.7 (31/92)	26.0 (58/223)	1.45	0.17
Resides in a rural location **^b^** (yes)	30.4 (28/92)	33.6 (75/223)	0.86	0.58
Lives in a house made of wood/leaves (yes)	44.6 (41/92)	48.9 (109/223)	0.84	0.49
Alcohol or tobacco consumption (yes)	55.4 (51/92)	51.6 (115/223)	1.17	0.53
**Comorbidity**, % (***n/N***)				
Diabetes or high blood pressure (yes)	6.5 (6/92)	7.6 (17/223)	0.85	0.73
Other cardiovascular disease (yes)	1.0 (1/92)	3.6 (8/223)	0.30	0.23
Digestive disease (yes)	8.7 (8/92)	5.4 (12/223)	1.68	0.27
Urinary disease (yes)	3.3 (3/92)	0.9 (2/223)	3.73	0.13
Dermatological disease (yes)	1.0 (1/92)	0.4 (1/223)	2.44	0.52
Other (yes)	0.0 (0/92)	0.9 (2/223)	1.42	0.36
**Previous infections**, % (***n/N***)				
Tuberculosis (yes)	20.7 (19/92)	22.0 (49/223)	0.92	0.80
** Intestinal parasitosis (yes)**	**18.5 (17/92)**	**9.0 (20/223)**	**2.30**	**0.017**
** Gonorrhea (yes)**	**19.6 (18/92)**	**10.3 (23/223)**	**2.12**	**0.026**
Syphilis (yes)	18.5 (17/92)	13.5 (30/223)	1.46	0.26
Chronic hepatitis (yes)	8.7 (8/92)	5.8 (13/223)	1.54	0.35
** Cerebral toxoplasmosis (yes)**	**0.0 (0/92)**	**4.9 (11/223)**	**1.43**	**0.038**
**Symptoms**, % (***n/N***)				
Cough, cold symptoms (yes)	14.1 (13/92)	8.1 (18/223)	1.87	0.10
Fever (yes)	1.1 (1/92)	2.2 (5/223)	0.48	0.68
Diarrhea (yes)	26.1 (24/92)	19.7 (44/223)	1.44	0.21
**Frequency of diarrhea, % (*n/N*)**				
No diarrhea	73.9 (68/92)	80.3 (179/223)	**N/A**	**0.014**
Once a month	16.3 (15/92)	16.6 (37/223)		
Once a week	2.2 (2/92)	2.2 (5/223)		
** Once a day**	**7.6 (7/92)**	**0.9 (2/223)**		
**Risk group**, % (***n/N***)				
Heterosexual	70.9 (61/86)	79.0 (166/210)	**N/A**	**0.046**
** Homosexual**	**27.9 (24/86)**	**15.7 (33/210)**		
Transexual/Bisexual	1.2 (1/86)	5.2 (11/210)		
■ **Missing data, % (** *n/N* **)**	6.5 (6/92)	5.8 (13/223)		
**HIV acquisition**, % ***(n/N***)				
Sexual	92.4 (85/92)	88.8 (198/223)	N/A	0.64
Vertical	0.0 (0/92)	0.9 (2/223)		
Parenteral	0.0 (0/92)	0.4 (1/223)		
Unknown	7.6 (7/92)	9.9 (22/223)		
**CD4+ nadir, median (IQR), /μL**	234 (131, 369)	261 (117, 378)	N/A	0.84
■ **Missing data, % (** *n/N* **)**	46.7 (43/92)	(92/223)		
**Current CD4+, median (IQR), /μL**	427 (265, 574)	431 (293, 592)	N/A	0.61
■ **Missing data, % (** *n/N* **)**	30.4 (28/92)	31.4 (70/223)		
**Current CD4+ < 200/mL, % (*n/N*), /mL**	**18.8 (12/64)**	**9.8 (15/153)**	**2.12**	**0.069**
■ **Missing data, % (** *n/N* **)**	30.4 (28/92)	31.4 (70/223)		
**Uncontrolled HIV viral load,** **(>20 copies/mL)**, % (***n/N***)	**29.4 (25/85)**	**18.4 (40/217)**	**1.84**	**0.037**
■ **Missing data, % (** *n/N* **)**	7.6 (7/92)	2.7 (6/223)		
**Poor ART adherence ≤ 95%, % (*n/N*)**	14.8 (12/81)	14.2 (26/183)	1.05	0.90
■ **Missing data, % (** *n/N* **)**	12.0 (11/92)	17.9 (40/223)		

Data are shown as % (*n/N*) or median (interquartile range: IQR), unless specified otherwise. Variables with a *p*-value < 0.10 are shown in bold and were included in the multivariable analysis. Percentages may not total 100 due to rounding. ^a^ Scientific work, teaching, architecture, or politics. ^b^ Defined as the absence of paved streets. Abbreviations: N/A: not applicable. ART: anti-retroviral therapy.

## Data Availability

The datasets used and/or analyzed during the current study are available in the Zenodo Repository, under the ORCID: 10.5281/zenodo.14864472.

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
