# Peer review of "Prevalence of Intestinal Protozoa Among Patients Living with HIV in the Peruvian Amazon"

_tropicalmed, 2025, doi:10.3390/tropicalmed10110324_

Round 1

Reviewer 1 Report

Comments and Suggestions for Authors

This manuscript presents a valuable and much-needed cross-sectional study on the prevalence of intestinal protozoa in people with HIV (PWH) in the understudied Peruvian Amazon region. The study's strengths include a substantial sample size, the use of multiple diagnostic techniques, and the identification of a key risk factor (homosexual practices). The finding of a very high prevalence of Cryptosporidium spp. (25.7%) is particularly noteworthy and has significant clinical and public health implications.

However, the manuscript requires major revisions before it can be considered for publication. The primary concerns are:

General comments

  1. There is significant confusion regarding the definition of "overall prevalence" for pathogens, particularly how results from different tests (Lugol's, MZN, ICT) were combined. This needs to be explicitly and transparently detailed.
  2. The stark discrepancy between MZN and ICT for Cryptosporidium is not adequately addressed. The authors must more deeply discuss the limitations of each method in this specific context, including the potential for MZN misidentification and the species-specificity of the ICT.
  3. The manuscript would benefit from careful proofreading to improve sentence structure, correct typographical errors, and ensure a logical flow between sections.
  4. There are several typographical errors (e.g., "resid in rural location" in Table 1, "E. hystolitica" in the Strengths and Limitations). A careful proofread is necessary.

Specific comments

Abstract

The abstract accurately summarizes the key findings. However, please specify that the "overall prevalence" for Cryptosporidium is a composite of MZN and ICT results, as this is a crucial methodological detail.

Materials and Methods

- The decision not to use a concentration technique (e.g., formalin-ether) is a significant limitation, as it reduces sensitivity for detecting low-burden infections, particularly for protozoan cysts. This should be acknowledged as a limitation in the discussion.

- The description of the MZN staining procedure is very detailed. While appreciated, it could be slightly condensed by referencing a standard protocol.

- It is good that you used Cohen's Kappa. Please confirm that the correct statistical test (e.g., McNemar's) was used to compare the diagnostic yields of the different tests, not just their agreement.

Results

 - The slight agreement (κ = 0.11) between MZN and ICT for Cryptosporidium is a critical finding. The cross-tabulation shows that 70 samples were positive by MZN but negative by ICT. This deserves a prominent and thorough discussion in the Discussion section.

- The flow chart (Figure 1) is confusing. The numbers at the top (208356) seem to be a typo. The chart should clearly show the initial number of patients approached, those who consented, and those with complete data.

- In table 1:

  • The title should specify that it presents bivariate analysis results.
  • The p-values for multi-category variables (e.g., Residence, Occupation, Education) are listed as "N/A" and then a single p-value is given. Please clarify which statistical test was used (e.g., Chi-square for trend?).
  • The footnote regarding missing data for HIV viral load (7.6% vs 2.7%) is concerning, as this imbalance could introduce bias. This should be mentioned in the limitations.

- The model was likely hampered by missing data (e.g., ~30% for CD4, ~5% for viral load). Please state the final number of participants included in the multivariate model and discuss the potential for reduced power in the limitations.

Discussion

- Cryptosporidium Diagnostic Dilemma: This section needs significant expansion. You must discuss the potential reasons for the MZN/ICT discrepancy in depth.

- MZN False Positives: Could the MZN-positive/ICT-negative results be due to misidentification of other acid-fast organisms (e.g., Cyclospora, Cystoisospora, yeast)? This is a well-known pitfall, especially in endemic areas.

- ICT Specificity: As you note, the ICT may only detect C. parvum. If C. hominis is the predominant species in your population (as in the Lima study you cite), the ICT would yield negative results. The high MZN prevalence may thus reflect the true burden of Cryptosporidium spp., but the possibility of MZN over-diagnosis cannot be ruled out without PCR confirmation.

- The finding regarding "homosexual practices" is important. Please discuss this in the context of the existing literature on sexual practices (e.g., oral-anal contact) as a route of transmission for enteric protozoa, linking it back to the fecal-oral pathway.

-The lack of a strong association between protozoa and diarrhea is interesting. Your suggestion of a multifactorial etiology (diet, other infections) is reasonable and should be elaborated.

- Please augment the limitations section to include: The lack of a concentration technique for stool samples, the inability to spectate Entamoeba or genotype Cryptosporidium, the potential for misclassification bias due to the MZN/ICT discrepancy, the impact of missing data on the statistical power of the multivariate analysis, and the cross-sectional design, which precludes causal inference.

Comments on the Quality of English Language

The manuscript would benefit from careful proofreading to improve sentence structure, correct typographical errors, and ensure a logical flow between sections.

There are several typographical errors (e.g., "resid in rural location" in Table 1, "E. hystolitica" in the Strengths and Limitations). A careful proofread is necessary.

Reviewer 2 Report

Comments and Suggestions for Authors

Dear Editor,

Thank you for the opportunity to review the manuscript entitled “Prevalence of Intestinal Protozoa among Patients Living with HIV in the Peruvian Amazon” by Otero-Rodriguez et al.

This is a well-conceived and informative study that contributes valuable data on intestinal protozoa infections in an underrepresented population. The findings are relevant for public health interventions in low-resource tropical settings. However, before acceptance, I recommend that the authors revise the manuscript.

  • The cross-sectional design is appropriate, but the sampling strategy should be better justified. Were patients enrolled consecutively or randomly? Clarify how representativeness of the HIV population in Iquitos was ensured.
  • The large difference between MZN and ICT results for Cryptosporidium (23.2% vs. 3.5%) is acknowledged, but further explanation or validation (e.g., inter-observer reliability, quality control measures) would strengthen confidence in the findings.
  • The logistic regression model should include clear criteria for variable selection.
  • Provide the number of cases excluded due to missing data to clarify potential bias in multivariable models.
  • While the limitations are well-described, the absence of molecular diagnostics is significant. Recommend expanding on how future studies could address this (e.g., genotyping or PCR validation).
  • Figure 2 lacks adequate labeling; include clearer legends and specify what “agreement” represents visually.
  • Minor grammatical errors should be corrected in final proofreading.
  • Line 25: Remove duplicated “Correspondence:”
  • Line 183: “CoxSnell” should be written “Cox & Snell”.
  • Abstract: Consider briefly mentioning that Blastocystis spp. was frequent but of uncertain pathogenicity.
  • Provide confidence intervals consistently in the Results section.

Reviewer 3 Report

Comments and Suggestions for Authors

Overall, I find the study well-built and informative.

I suggest some minor reviews, as follows:

  • Lines 116-119: I guess samples were stored at room temperature in patients' houses. Do we know how long samples remained at room temperature before being collected by drivers, on average (and at the most)? How much did it take for drivers to take samples to the clinic, once they had collected them? These are logistic difficulties that are perfectly understandable in this setting, however I would specificy it.
  • Lines 130-132: this is perfectly understandable considering the logistic/management issues that likely existed during the study. However, to concentrate samples before Lugol's iodine solution is still adviced, despite some reports questioning its real utility. Therefore, I wouldn't justify this only with some reports showing a non-significant difference in sensitivity between direct examined and concentrated samples, but also discussing the feasibility to perform concentration methods in the context of this study (if this was the reason why it was not done).
  • Lines 136-137: phagocytosed erythrocytes may be observed even in a small minority of cases of Entamoeba dispar, I would remark it is not a pathognomonic sign.
  • Line 198: are the numbers in brackets IQR or SD?
  • Line 204: I would report the absolute percentage of subjects with diarrhea once a month, once a week etc., not the percentage relative to subjects with diarrhea only.
  • Line 205: you mean before and after excluding the 41 patients without MZN/ICT? Please clarify.

Probably, the main issue of the study is the data about Cryptosporidium, due to the big inconsistency between the results of microscopy and the ones of ICT. However, the authors comment about this in the discussion, and the two possible explanations they give are the same that I thought.
